# Antisense Therapy for Infectious Diseases

**DOI:** 10.3390/cells12162119

**Published:** 2023-08-21

**Authors:** Lwanda Abonga Buthelezi, Shandre Pillay, Noxolo Nokukhanya Ntuli, Lorna Gcanga, Reto Guler

**Affiliations:** 1International Centre for Genetic Engineering and Biotechnology, Cape Town Component, Cape Town 7925, South Africa; ndwlwa002@myuct.ac.za (L.A.B.); shandre.pillay@uct.ac.za (S.P.); ntlnox001@myuct.ac.za (N.N.N.); lornagcanga@yahoo.com (L.G.); 2Department of Pathology, Division of Immunology, Institute of Infectious Diseases and Molecular Medicine (IDM), Faculty of Health Sciences, University of Cape Town, Cape Town 7925, South Africa; 3Faculty of Health Sciences, Wellcome Centre for Infectious Diseases Research in Africa, Institute of Infectious Diseases and Molecular Medicine, University of Cape Town, Cape Town 7925, South Africa

**Keywords:** antisense oligonucleotide, antisense therapy, mRNA, infectious disease

## Abstract

Infectious diseases, particularly Tuberculosis (TB) caused by *Mycobacterium tuberculosis*, pose a significant global health challenge, with 1.6 million reported deaths in 2021, making it the most fatal disease caused by a single infectious agent. The rise of drug-resistant infectious diseases adds to the urgency of finding effective and safe intervention therapies. Antisense therapy uses antisense oligonucleotides (ASOs) that are short, chemically modified, single-stranded deoxyribonucleotide molecules complementary to their mRNA target. Due to their designed target specificity and inhibition of a disease-causing gene at the mRNA level, antisense therapy has gained interest as a potential therapeutic approach. This type of therapy is currently utilized in numerous diseases, such as cancer and genetic disorders. Currently, there are limited but steadily increasing studies available that report on the use of ASOs as treatment for infectious diseases. This review explores the sustainability of FDA-approved and preclinically tested ASOs as a treatment for infectious diseases and the adaptability of ASOs for chemical modifications resulting in reduced side effects with improved drug delivery; thus, highlighting the potential therapeutic uses of ASOs for treating infectious diseases.

## 1. Introduction

Antisense therapy involves administering antisense oligonucleotides (ASOs) to targets and inhibiting the translation of disease-associated messenger RNA (mRNA) [1]. ASOs are short, single-stranded, synthetic polymer nucleotides that regulate gene expression by gene silencing or modulating splicing [2]. While they hinder protein expression, they can also promote the expression of desired proteins through splice modulation of targeted pre-mRNA [3,4]. These genetically engineered oligonucleotides specifically bind to complementary target pre-mRNA or mRNA [5].

ASOs can be categorized into two broad modes of action based on their design: RNA cleavage and RNA blockage. ASO-mediated RNA cleavage involves molecular RNA degradation using ribonuclease RNAse-1-H [6] or gene silencing through RNA interference (RNAi) [7]. ASO-mediated RNA blockage relies on steric hindrance achieved by advanced binding affinity or splice modulation, resulting in exon exclusion [8] or inclusion [9], depending on the desired outcome [10,11]. Compared to conventional therapies, antisense therapies have shown promising success due to their engineered target specificity, leading to fewer side effects like cytotoxicity. Chemical modifications have led to improved generations of ASOs, enhancing resistance to degradation, binding affinity, delivery, cellular uptake, and intracellular trafficking [12,13,14,15]. The efficacy of ASOs was improved by introducing chemical modifications to the ribose sugar or phosphorothioate (PS) backbone of the molecule [16]. These chemical modifications further provide the ability to customize the ASOs for different clinical applications.

The major challenge with current antimicrobial therapies for infectious diseases is the emergence of multi-drug-resistant pathogens, as seen in tuberculosis, the leading cause of global death due to drug-resistant strains [17]. Therefore, the search for highly effective, safe, and precise therapies is crucial for global health development [18]. Antisense therapies have demonstrated their potential in inhibiting viral replication and silencing disease-associated genes [19,20], making them a viable option for emerging and re-emerging infectious diseases. ASOs can be employed as host-directed therapy adjuncts to existing treatments, targeting host factors contributing to pathogenesis [21,22]. This not only enhances the host’s immune response but also reduces drug resistance. ASO-based therapies hold promise for personalized therapy and may revolutionize disease treatment [23].

## 2. ASO Modifications and Delivery

Previously, chemically unmodified ASOs were ineffective as RNA therapeutics due to their larger size, charge, and the presence of the phosphodiester bond. These characteristics hindered the passive diffusion of ASOs into the target cell [24], rendering them susceptible to degradation by nucleases [25]. However, newer generations of ASOs have been developed to address these limitations, offering enhanced efficiency, accuracy, reduced toxicity, and enzymatic stability. The first significant chemical modification involved the introduction of phosphorothioate (PS) linkages, wherein a phosphate group replaced the non-bridging oxygen with a sulfur group [26]. PS-ASOs exhibit improved enzymatic stability, effectively resisting nuclease degradation [27] and enhancing efficiency through targeted mRNA degradation by RNase H [28]. Despite these improvements, the first-generation ASOs were associated with some adverse effects, such as low binding affinity and immune stimulation [29].

The second generation of ASOs includes 2′-modified phosphorothioates, such as 2′-O-methyl (2′-O-Me) and 2′-O-methoxyethyl (2′-O-MOE) oligonucleotides, which have alkyl modifications at the 2′-position of the ribose [30]. These ASOs are highly resistant to nuclease degradation and have higher target binding affinity [31,32]. For instance, IONIS-HBV, a 2′-O-Me ASO, is currently undergoing clinical trials for chronic hepatitis B (HBV) infection. Phase 2 findings revealed a dose-dependent reduction in HBV DNA and the HBV surface antigens [33].

The third generation comprises Phosphorodiamidate Morpholino Oligomers (PMOs) with morpholine rings as a backbone connected with Phosphorodiamidate linkages [34], Peptide Nucleic Acids (PNAs) replacing the ribose phosphate backbone with N-aminoethylglycine polyamide [35], and Locked Nucleic Acids (LNAs) containing a methylene bridge between 2′-O and 4′-C [36]. These recent chemical modifications have produced improved ASOs with overall higher enzymatic stability, increased binding affinity, reduced immune stimulation, and better pharmacokinetics [37]. Notably, one of the third-generation ASOs currently undergoing clinical trials is Miravirsen, a LNA-ASO targeting miR-122 in the context of hepatitis C (HCV) [38]. Efficient cellular uptake of ASOs can be achieved by conjugating them to charged molecules like N-acetylgalactosamine (GalNAc) [39,40] or using cell-penetrating peptides (CPP) designed for selective drug delivery [41]. Another approach involves using unmethylated cytosine-guanine dinucleotide (CpG) oligodeoxynucleotides (CpG-ODN), which trigger an innate immune response by binding to Toll-like receptor 9 (TLR9) [42].

Drug delivery to the target site poses a significant challenge in ASO therapies, primarily because ASOs must surmount biological barriers, escape lysosomal degradation, and avoid becoming trapped in secretory vesicles [16]. To address this issue, various strategies are currently being developed to enhance ASO stability and trafficking. As discussed, chemical modifications have proven effective in improving ASOs’ resistance to nucleases, binding affinity, cell penetration, and reducing off-target effects, thus enhancing the delivery of therapeutic oligonucleotides. One promising approach to advancing drug delivery involves conjugating ASOs with peptides, aptamers, antibodies, and N-acetylgalatosamine (GalNAc). Peptide conjugates, for instance, can traverse cell membranes and induce endosome disruption for efficient drug release [43]. Antibody conjugates, on the other hand, facilitate the internalization of ASO drugs, increasing their bioavailability in target organs [40]. Aptamers, by binding to cell surface proteins and mimicking antibody function [44], offer another avenue to improve ASO delivery. Moreover, GalNAc conjugation enhances ASO potency and provides a reliable entry route to hepatocytes [45]. In clinical trials targeting chronic HBV infection, Bepirovirsen was conjugated to GSK3389404, a GalNAc conjugate, with the aim of enhancing drug delivery to hepatocytes [46].

In addition to GalNAc conjugation, researchers are also exploring nano-drug vehicles, such as liposomes, for ASO delivery [47]. Liposomes consist of a lipid bilayer encapsulating an aqueous compartment and have shown great promise for drug delivery due to their ability to accumulate in diseased cell populations [48]. Liposomes have been extensively researched and tested in clinical trials due to their stability, selective delivery capacity, low toxicity, biocompatibility, and potential for conjugation with targeting moieties like peptides for cell-specific drug delivery [16,49]. These advancements in drug delivery technologies hold significant potential for improving the efficacy and safety of ASO therapies in various diseases.

## 3. Mode of Action

The use of ASOs as a treatment for diseases offers significant advantages in therapeutics. ASOs can be specifically modified and applied for different diseases or unique genetic disorders, such as Milasen, which is tailored to individual patients [50]. ASOs function through the Watson–Crick base pairing method, specifically binding to their cognate target pre-mRNA or mRNA [51]. ASOs are classified based on their mode of action and their dependence on the RNA-degrading enzyme RNase H. They are categorized as either RNA-cleaving/degrading ASOs or RNA-blocking/steric-hindrance ASOs, also known as RNase H-dependent and RNase H-independent ASOs, respectively [52].

### 3.1. RNA Degrading ASOs

The RNA-degrading ASOs rely on RNase H-mediated RNA degradation, involving the binding of the target mRNA’s RNA-DNA heteroduplex to the complementary ASO [53]. This recruits the RNase H1 enzyme, which degrades the RNA in the heteroduplex. Consequently, the translation of the target mRNA is inhibited, leading to the downregulation of the corresponding protein [54,55]. FDA-approved ASOs that utilize this method include fomivirsen, mipomersen, and inotersen [56].

### 3.2. RNA Blocking ASOs

RNA-blocking ASOs have been primarily used as a treatment for genetic disorders like Duchenne Muscular Dystrophy (DMD). In this approach, ASOs repair the reading frame, resulting in the production of a functional protein [57]. ASOs’ advanced binding affinity is crucial in this process, as they directly bind to the specific targeted pre-mRNA or mRNA. Recently, FDA-approved ASOs have been combined with alternative splicing mechanisms, such as exon skipping and exon inclusion, for disease treatment. Exon skipping involves the ASO binding to the targeted exon in the pre-mRNA, leading to its exclusion during translation and inducing the expression of a shorter yet functional protein. This method has been applied in eteplirsen [58], golodirsen [59], and casimersen for DMD treatment [60]. On the other hand, exon inclusion restores the full length of the mRNA, resulting in the upregulation of the desired functional protein [61]. Nusinersen employs this method for the treatment of SMA [56].

## 4. FDA-Approved ASOs

Vitravene, also known as fomivirsen, was the pioneering ASO-approved drug, authorized in 1998 for the treatment of human cytomegalovirus (HCMV) retinitis [62]. HCMV infection can lead to vision loss, particularly in immunocompromised AIDS patients [63]. Vitravene, a 21-mer phosphorothioate oligo-2′-deoxynucleotide, was designed to be complementary to its target HCMV mRNA, where it binds to the major immediate early region. By doing so, Vitravene reduces HCMV viral replication and decreases viral load [64,65]. Vitravene treatment showed no extreme adverse effects, although some patients reported intraocular inflammation and increased pressure. However, these symptoms were not directly linked to Vitravene but rather to HCMV infection [66,67].

## 5. Therapeutic Applications

Antimicrobial oligonucleotides target the pathogens’ essential genes responsible for replication, antibiotic resistance, and virulence. By targeting these genes, it is possible to augment the host’s immune response, resulting in reduced pathogen replication and proliferation.

### 5.1. Antibacterial Oligonucleotides

As of now, there are no FDA-approved antibacterial oligonucleotides available. However, preclinical studies have shown promising results in exploring ASOs as potential antibacterial agents [68]. Notably, back in 1981, it was demonstrated that oligonucleosides methylphosphonates could effectively inhibit the growth of *Escherichia coli* [69]. Additionally, lipid oligonucleotides have been utilized to efficiently deliver the oligonucleotide sequence, thereby decreasing the Minimum Inhibitory Concentration (MIC) and effectively combating antibiotic resistance [70].

Pathogens have acquired resistance to the currently available antibacterial therapies, rendering them ineffective. The possibility of using antisense therapy as an adjunct therapy has been demonstrated in culture, where methicillin-resistant Staphylococcus aureus was sensitized using PNAs and PS-ODNs. These strains were then found to be susceptible to oxacillin, decreasing bacterial growth [71]. Another challenge with bacterial infections is the formation of biofilms, which can enhance bacterial virulence due to the overexpression of non-essential genes coding for bacterial virulence, necessitating a larger dose of treatment. Targeting the genes involved in biofilm formation and inhibition was previously discussed as advances in bacterial antisense therapy [72] and has shown effectiveness against *Haemophilus influenzae* [73] and *Pseudomonas aeruginosa* [74]. Table 1 outlines chemically modified ASOs and their bacterial target genes as antisense therapy for bacterial infections.

The peptide-conjugated PNAs’ 1 and 2 (PPNA1 and PPNA2) ASOs are complementary to different sites of the target filamentous temperature-sensitive protein Z (ftsZ) gene, which is critical for the replication of methicillin-resistant *Staphylococcus aureus* (MRSA) [75]. Both PPNA1 ASO and PPNA2 ASO displayed bactericidal effects and inhibited the growth of MRSA in cell culture [75] (Figure 1A1). The peptide-conjugated LNA, PLNA787 ASO, targets the ftsZ mRNA of *Staphylococcus aureus* [76]. Treatment with PLNA787 ASO treatment inhibited the growth of methicillin-resistant *Staphylococcus aureus* in cell cultures. Moreover, its therapeutic impact extended to in vivo studies, as evidenced by an increased survival rate among mice infected with the Mu50 strain of *S. aureus* [76] (Figure 1A2). Furthermore, in a *Staphylococcus aureus*-infected cutaneous mouse wound, the use of PMO conjugates in conjunction with a novel thermoresponsive gel delivery system targeted the essential gyrA mRNA, resulting in decreased bacterial growth and enhanced cutaneous wound healing in the mice [77] (Figure 1A3). The rpoA-PNA ASO, a CPP-PNA, effectively targeted the RNA polymerase α subunit (rpoA) involved in transcription in *Listeria monocytogenes* [78]. By inhibiting bacterial DNA transcription, this rpoA-PNA ASO significantly reduced bacterial growth of *L. monocytogenes* in broth culture. In vivo treatment of *L. monocytogenes*-infected *Caenorhabditis elegans* worms with rpoA-PNA resulted in a remarkable 72% growth reduction [78]. (Figure 1B). Similarly, the polA-PNA ASO, a CPP-PNA, effectively targeted the DNA polymerase I (polA) gene responsible for DNA replication in *Brucella suis* [79]. Treatment with 12–30 μM of polA-PNA ASO inhibited the growth of *Brucella suis* in both broth culture and infected macrophages [79] (Figure 1C). Additionally, the acpP-PPMO ASO targeted the acyl carrier protein (acpP) gene involved in the fatty acid biosynthesis of *Acinetobacter baumannii.* Intranasal treatment with acpP-PPMO ASO demonstrated its bactericidal effect, significantly increasing the survival rate of *A. baumannii*-infected mice while reducing inflammation and bacterial burden in the lungs [80] (Figure 1D).

### 5.2. Antiviral Oligonucleotides

Antiviral oligonucleotides have been extensively studied in preclinical research as a therapeutic approach to target specific virus-specific mRNAs. Notably, previous studies have explored the potential applications of antiviral oligonucleotides in combating viral infections, including Dengue fever. For instance, the peptide-conjugated phosphorodiamidate morpholino oligomer (PPMO) ASO effectively inhibited Dengue virus replication by blocking RNA translation and RNA synthesis [81]. Various ASOs have been employed to treat different viral infections, leading to a notable overall decrease in viral titers, as summarized in Table 2.

A 3′-SLT PPMO ASO was designed to target the 3′ stem-loop (3′SLT) in the Dengue viral genome imperative for the translation and synthesis of viral RNA [88,89]. The 3′SLT PPMO effectively reduced viral RNA levels by over 450-fold, thereby inhibiting viral translation and RNA synthesis in Baby Hamster Kidney fibroblasts (BHK) cells [81] (Figure 2A1). Another ASO, Vivo-MO-1, effectively targeted the 3′ stem-loop (3′ SLT) on the 3′ UTR of the Dengue viral genome, resulting in the inhibition of Dengue infection by decreasing viral RNA levels by over 1000-fold in dendritic cells treated with ASO Vivo-MO-1 [82] (Figure 2A2). In the context of Hepatitis B virus (HBV) infection, an LNA-single-stranded oligonucleotide (LNA-SSO) conjugated to N-acetylgalactosamine (GalNAc) ASO was utilized to target HBV transcripts in human hepatoma cell lines. The conjugation to GalNAc aids in the specific binding to asialoglycoprotein receptor (ASGPR), which is expressed on hepatocytes preventing the accumulation of ASO in the kidney. This led to a long-lasting reduced expression of viral antigens (HBsAg and HBeAg) and mRNA [86] (Figure 2B). Furthermore, promising results were achieved in treating Ebola infection in mice using a PMO ASO conjugated with an arginine-rich peptide (PPMO). This PPMO ASO effectively targeted the VP24 mRNA and inhibited viral replication [85]. Notably, treatment with 50 μg and 5 μg of VP24-AUG PPMO resulted in 100% and 90% protection against lethal Ebola infection in mice, respectively [85]. (Figure 2C). Another successful example involves the P7-PMO ASO, which targeted the 3′-terminal region of nucleoprotein viral genome RNA (NP-v3′), displaying enhanced antiviral activity with a >85% reduction in viral Influenza titers in infected Madin–Darby canine kidney (MDCK) cells [84] (Figure 2D). Another successful example involved the AUG-2 PPMO ASO, designed to target the translational start site of the RSV-L mRNA. This ASO showed significant efficacy with a >2.0 log10 decrease in RSV viral titers in infected mouse and human cell lines [83] (Figure 2E). Lastly, the 2′-deoxy-2′-fluoroarabinonucleotide (FANA) ASO effectively targeted the conserved regions of the HIV-1 genome, resulting in the inhibition of viral p24 replication in human PBMCs. They further investigated the duration of the ASOs’ anti-HIV effects and noted that treatment with FANA ASO had a prolonged viral p24 inhibition period of 13 days [87] (Figure 2F). Finally, the 2′-deoxy-2′-fluoroarabinonucleotide (FANA) ASO effectively targeted the conserved regions of the HIV-1 genome, resulting in the inhibition of viral p24 replication in human PBMCs. Investigations into the longevity of the ASO’s anti-HIV effects revealed an impressive duration, with treatment using FANA ASO resulting in the prolonged inhibition of viral p24 for a period of 13 days [87] (Figure 2F).

### 5.3. Antiparasitic Oligonucleotides

Parasitic infections represent one of the most neglected types of infectious diseases due to the lack of improved and updated therapies. Currently, there are no FDA-approved antisense therapies specifically targeting parasitic infections, nor are there any ASOs in clinical trials for such purposes. However, preclinical studies have shown promising results using oligonucleotides to target parasitic infections, as summarized in Table 3. An excellent example of the potential of antiparasitic applications in preclinical studies is demonstrated by PPMOs and VMOs that target essential genes of *Plasmodium falciparum*, leading to reduced RNA expression and inhibition of parasite growth [90]. Additionally, oligonucleotide therapy can be utilized to target genes responsible for drug resistance in the parasite. This approach was successful in treating chloroquine-resistant *Plasmodium falciparum*, where a MO conjugate was used, resulting in restored chloroquine susceptibility [90]. Table 3 outlines the various oligonucleotides and their target genes for different parasitic infections.

For instance, GRA10-PPMO ASO targets the GRA10 mRNA, resulting in the downregulation of the GRA10 granular protein and disrupting the intracellular replication of *Toxoplasma gondii* in human fibroblasts [91] (Figure 3A). Furthermore, PfCRT-VMO ASO (1) and PfDXR-PPMO ASO (2) are designed to target essential genes, PfCRT is responsible for chloroquine resistance, and PfDXR is involved in apicoplast formation, which is crucial for metabolic functions in *Plasmodium falciparum* [90]. Inhibiting the expression of the PfCRT gene resulted in restored drug susceptibility, and treatment with 1.25 µM and 1.75 µM of PfPMT-VMO and PfCRT-VMO, respectively, inhibited 53% of parasitic growth in the presence of chloroquine [90] (Figure 3B1,2). Another approach involves the PNA ASO, which targets PfSec13, a gene involved in *Plasmodium falciparum* proliferation. The downregulation of PfSec13 leads to decreased parasitic proliferation in human erythrocytes [93] (Figure 3B3). Similarly, the antisense 5995 ASO targets the inositol 1,4,5-trisphosphate receptor (TcIP3R) of *Trypanosoma cruzi*, a gene that plays a role in the parasite’s replication and virulence [92]. By treating trypomastigotes, the infective stage of *T. cruzi*, reduced TcIP3R expression hampers the parasite’s ability to invade cells and replicate [92] (Figure 3C).

## 6. Anti-Mycobacterial and Host Factor Targeted ASOs for *Mycobacterium Species*

Several ASOs have demonstrated successful inhibition of disease-causing genes during Mycobacterium infections through knockdown experiments. For instance, the knockdown of an immunoregulatory long intergenic noncoding RNA, lincRNA-MIR99AHG, using an LNA GapmeR-ASO, significantly reduced the intracellular growth of *Mycobacterium tuberculosis* in murine and human macrophages [94]. In mice, intranasal administration of LNA GapmeR-ASO led to a substantial reduction in bacterial burden within the lungs (Figure 4A) [94]. Moreover, a phosphorothioate-modified antisense oligodeoxyribonucleotide (PS-ODN) was found to inhibit the expression of inositol-1-phosphate synthase, a key enzyme encoded by the inositol-1 (INO1) gene, in inositol synthesis [95]. This suppression resulted in a notable decrease in mycothiol levels following exposure to 20 and 40 μM of the PS-ODN over a span of 6 weeks, consequently reducing the proliferation of *M. tuberculosis*. The ASO PS-ODN’s impact was remarkable, rendering the bacteria 7–9 times more susceptible to antibiotics at the 40 μM concentration (Figure 4B) [96]. Additionally, phosphoryl guanidine oligo-2′-O-methylribonucleotides (2′-OMe PGOs) have demonstrated successful penetration of *Mycobacterium smegmatis* and inhibited the expression of alanine dehydrogenase, encoded by the mycobacterial ald gene. Consequently, treatment with 20 μM of these PGOs resulted in a substantial reduction in *M. smegmatis* growth by 54% and 62% at 24 and 40 h, respectively, in murine macrophages (Figure 4C) [97]. These findings showcase the versatility of ASOs as potent therapeutic agents, offering multiple mycobacterial and host-directed targeting for combatting TB.

## 7. ASOs Targeting Host Factors for Viral Infections

Investigating potential ASO therapy targets for infectious diseases has led to a focus on host factors that viral pathogens manipulate for their survival. The ability of ASOs to disrupt viral pathogenesis by silencing essential host factors has shown great potential in inhibiting viral replication and propagation, providing new possibilities for the treatment of infectious diseases.

ASOs can silence viral targets using steric hindrance, thereby disrupting viral pathogenesis and rendering the virus susceptible to the host’s immune response. Figure 5 illustrates ASOs labelled A–F, which target different host factors responsible for viral replication and propagation. In the case of SARS-CoV-2 infection, the viral spike protein attaches to the angiotensin-converting enzyme 2 (ACE2) receptor on the host cell, enabling cellular entry [98]. The ACE2 ASO targets the ACE2 gene, resulting in reduced expression and preventing SARS-CoV-2 cellular entry and release of the viral genome [99] (Figure 5A). MicroRNA-122 (miR-122) plays a role in the replication of the Hepatitis C virus (HCV) in infected human Huh7 cells. The 2′-O-methylated RNA oligonucleotide (122-2′OMe) targets miR-122, preventing the stabilization of the viral genome and hindering viral replication [100] (Figure 5B). For Ebola virus, binding to Niemann–Pick C1 (NPC1) mRNA prompts the release of the ribonucleoprotein (RNP) complex responsible for viral transcription and replication [101]. An LNA ASO that directly binds to the NPC1 mRNA inhibits membrane fusion, prevents RNP release, and reduces viral replication in Ebola-infected murine and human cell lines [102] (Figure 5C). In the case of H1N1 influenza, binding to the programmed cell death protein 5 (PDCD5) induces apoptosis and viral replication [103]. PROP5 ASO binds to the PDCD5 mRNA and impedes viral propagation of H1N1 influenza [103] (Figure 5D). Hepatitis B virus (HBV) replication has been associated with the virus binding to asialoglycoprotein receptor 1 (ASGPR1). ASODN2 ASO binds to ASGPR1 mRNA, resulting in the inhibition of HBV replication [104] (Figure 5E). The host factor Mammalian relative of DnaJ (MRJ/DNAJB6) is involved in both HIV-1 and respiratory syncytial virus (RSV) infection. For HIV-1, MRJ promotes viral entry into the host nucleus [100], and for RSV, it is involved in the production of viral subgenomic RNA [105]. The binding of MoMRJ ASO to the MRJ-L isoform suppresses HIV-1′s nuclear entry and viral genome integration and inhibits RSV’s viral RNA and mRNA expression [105,106] (Figure 5F).

## 8. Advantages and Disadvantages

### 8.1. Advantages

Antisense therapy has revolutionized the outlook for future treatments in patients with various diseases, offering the potential for personalized and accessible treatment options [103]. One major advantage of this therapy lies in its simplicity and accuracy, as it is complementary-based and requires the target mRNA sequence. This allows for the synthesis of specific ASOs targeting known sequences of viral or bacterial mRNAs involved in pathogenesis [107]. This approach has been particularly valuable in cases where protein-based therapies were ineffective due to the inaccessibility of the disease target. Chemical modifications to oligonucleotide structures have significantly improved therapy efficacy and safety by enhancing binding affinity and reducing toxicities. Moreover, the production time and cost-effectiveness of ASO therapy are advantageous, especially for large-scale production, while its increased efficacy reduces the need for frequent drug administration.

### 8.2. Disadvantages and Improvements

Despite its promising potential, the clinical use of oligonucleotides has faced challenges. The delivery of first-generation oligonucleotides to specific organs and tissues has been problematic due to their weight, negative charge, and instability, resulting in poor absorption and susceptibility to nuclease degradation [108]. However, second- and third-generation ASOs with chemical modifications, including cell-penetrating peptides, have addressed these issues and facilitated delivery to target organs. Concerns about off-target toxicity observed in first-generation oligonucleotides [109] have been minimized in later generations due to improved chemical modifications, leading to reduced toxicity. Moreover, specific oligonucleotide therapies, like small interfering RNA, have been linked to heightened immune responses, leading to inflammatory syndromes. This phenomenon was primarily observed in therapies using small interfering RNA that acted as TLR agonists, resulting in clinical adverse effects [110].

## 9. Conclusions and Future Perspectives

Antisense therapy has undergone significant evolution over the years, with numerous advancements aimed at enhancing its efficacy and applications in the field of health sciences. The ability to modify protein expression, either by reducing or restoring it without toxicity, has presented remarkable therapeutic possibilities for numerous diseases and genetic disorders. Improved chemical modifications of ASOs have effectively addressed many concerns associated with earlier first-generation oligonucleotide therapies. From the introduction of the phosphorothioate linkage, which improved enzymatic stability, to the conjugation of cell-penetrating peptides for enhanced cellular uptake, ASOs have become more versatile and potent in their therapeutic potential.

The applications of ASOs in infectious diseases have been extensively explored and discussed in this review. Most of the antimicrobials discussed have been rigorously tested in cell cultures, mouse models, and human cell lines. Notably, one significant finding in ASO applications is their potential as host-directed therapy. By targeting multiple host factors involved in viral entry and replication, complementary ASOs can effectively inhibit viral replication and growth.

ASOs offer promising prospects for actively combating emerging, re-emerging, and multi-drug-resistant infectious diseases. However, there is still ample room for improvement in most antisense therapeutics. Optimizing ASO delivery to specific target organs can prevent undesirable accumulation in the kidney and liver, thus enhancing overall efficacy. Novel delivery modalities for oligonucleotide therapies can further improve their effectiveness while reducing the required drug dosage. Leveraging approved delivery strategies used in other diseases should be thoroughly explored as they may prove advantageous.

In conclusion, ASO therapies hold tremendous potential for treating various diseases, including those caused by bacteria, viruses, and parasites. To fully harness this potential, addressing drug delivery challenges remains a key focus. Through ongoing research, preclinical studies, and clinical trials, the optimization of ASO design, delivery, and personalized medicine can revolutionize the treatment landscape, benefiting patients worldwide and offering new hope in the fight against infectious diseases.

## Figures and Tables

**Figure 1 cells-12-02119-f001:**
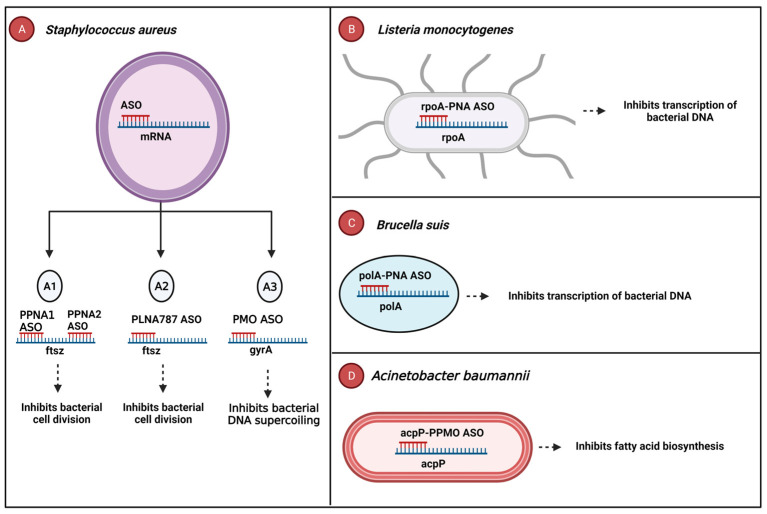
Antibacterial oligonucleotides targeting essential bacterial genes. (**A1**) PPNA1/2 ASOs target ftsZ mRNA and inhibit the growth of methicillin-resistant *Staphylococcus aureus.* (**A2**) PLNA787 ASO binds to ftsZ mRNA of *Staphylococcus aureus* to inhibit its growth. (**A3**) PMO ASO attaches to gyrA mRNA, causing a growth reduction in *Staphylococcus aureus*. (**B**) rpoA-PNA ASO binds to rpoA mRNA resulting in a reduction in *L. monocytogenes* growth. (**C**) polA-PNA ASO attaches to the polA mRNA, inhibiting the growth of *Brucella suis.* (**D**) The binding of acpP-PPMO ASO with the acpP mRNA results in the reduction in *Acinetobacter baumannii* growth. Abbreviations: ftsZ, filamentous temperature-sensitive protein Z; gyrA, gyrase A; rpoA, RNA polymerase α subunit; polA, DNA polymerase I; acpP, acyl carrier protein. Created in BioRender.com.

**Figure 2 cells-12-02119-f002:**
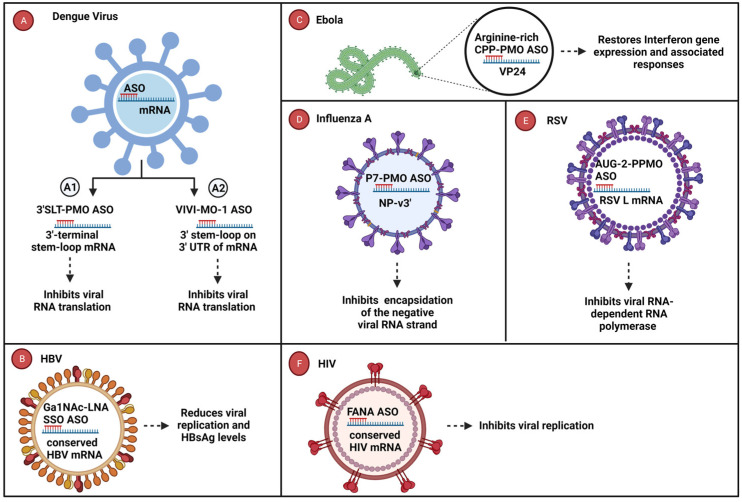
Antiviral oligonucleotides targeting essential viral genes. (**A1**) 3′SLT PPMO ASO targets the 3′ terminal stem-loop of the Dengue viral genome, inhibiting viral translation and RNA synthesis. (**A2**) Vivo-MO-1 ASO binds to the 3′ terminal stem-loop of the Dengue viral genome and inhibits the production of viral proteins. (**B**) Ga1Nac-LNA (SSO) ASO binds to the hepatitis B virus (HBV) transcript preventing the expression of viral antigens. (**C**) PMO ASO conjugated to an arginine-rich peptide (PPMO) successfully inhibited the viral replication of Ebola by attaching to the VP24 mRNA. (**D**) P7-PMO ASO targets the NP-v3′ site on the viral mRNA, obstructing the replication of Ebola. (**E**) AUG-2 PPMO ASO binds to respiratory syncytial virus (RSV)-L mRNA leading to reduced viral titers. (**F**) FANA ASO binds to the conserved regions of the HIV-1 genome, inhibiting HIV-1 replication. Abbreviations: 3′SLT, 3′ stem loop; Ga1NAc, N-acetylgalactosamine; NP-v3′, nucleoprotein viral genome; FANA, 2′-deoxy-2′-fluoroarabinonucleotide; HBV, Hepatitis B virus; RSV, Respiratory syncytial virus; HIV, Human immunodeficiency virus. Created in BioRender.com.

**Figure 3 cells-12-02119-f003:**
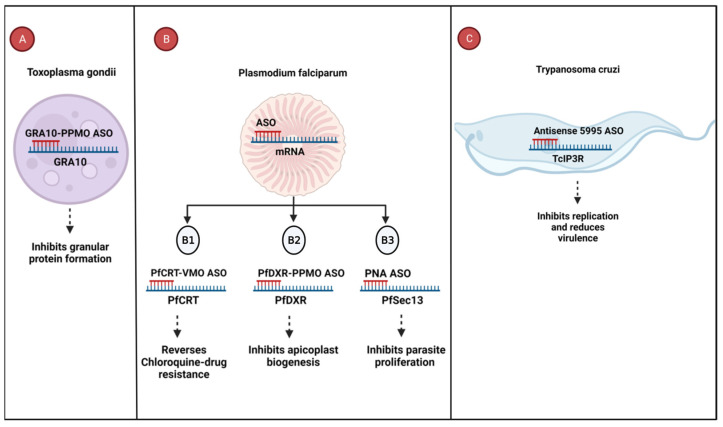
Antiparasitic oligonucleotides targeting essential parasitic genes. (**A**) GRA10-PPMO ASO downregulates GRA10 expression, resulting in growth inhibition *Toxoplasma gondii*. (**B1**) PfCRT-VMO ASO and (**B2**) PfDXR-PPMO ASO target the essential genes, PfCRT and PfDXR, respectively, leading to growth inhibition of *Plasmodium falciparum.* (**B3**) The binding of PNA ASO and PfSec13 results in decreased *P. falciparum* proliferation. (**C**) Antisense 5995 ASO binds to the TcIP3R mRNA of *T. cruzi*, inhibiting cell entry and replication. Abbreviations: GRA10, dense granule protein 10; PfCRT, *P. falciparum* chloroquine resistance transporter; PfDXR, *P. falciparum* deoxyxylulose 5-phosphate reductoisomerase; TcIP3R, *T. cruzi* inositol 1,4,5-trisphosphate receptor. Created in BioRender.com.

**Figure 4 cells-12-02119-f004:**
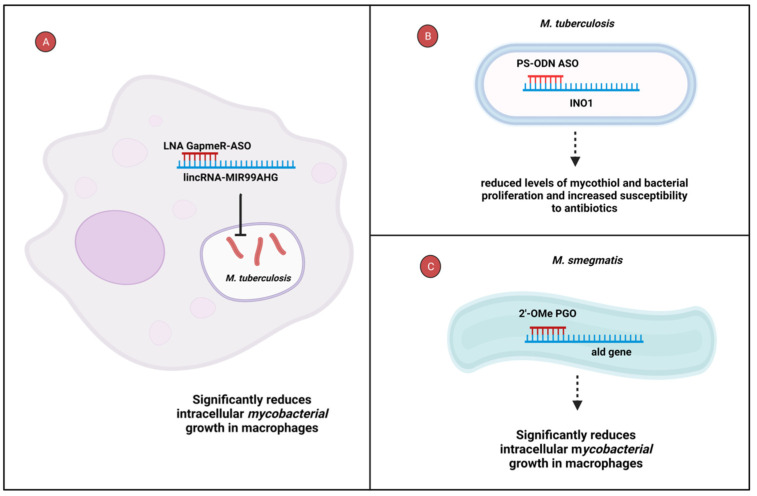
ASOs targeting host factors and essential bacterial genes. (**A**) The LNA GapmeR-ASO binds to the host lincRNA-MIR99AHG, resulting in the reduction in *Mycobacterium tuberculosis* in both murine and human macrophages. (**B**) PS-ODN ASO effectively targets the INO1 gene, resulting in reduced mycothiol levels and suppressed proliferation of *M. tuberculosis.* This, in turn, enhances the bacteria’s susceptibility to antibiotics (**C**) The 2′-OMe PGOs target the mycobacterial ald gene, leading to a significant inhibition of *M. smegmatis* growth within murine macrophages: LNA, locked nucleic acid; lincRNA-MIR99AHG, long intergenic noncoding RNA-MIR99AHG; PS-ODN, phosphorothioate-modified antisense oligodeoxyribonucleotide; INO1, inositol-1; *M. tuberculosis*, *Mycobacterium tuberculosis*; 2′-OMe PGOs, phosphoryl guanidine oligo-2′-O-methylribonucleotides; *M. smegmatis, Mycobacterium smegmatis.* Created in BioRender.com.

**Figure 5 cells-12-02119-f005:**
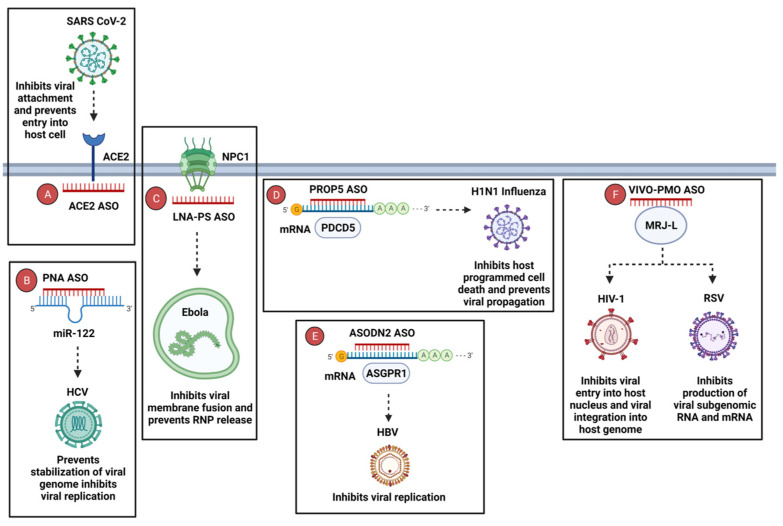
ASOs targeting host factors during various viral infections in the host cell. (**A**) ACE2 ASO binds to the ACE2 receptor, inhibiting the entry of SARS-CoV-2. (**B**) PNA ASO attaches to miR-122, preventing HCV viral genome stabilization and replication. (**C**) LNA-PS ASO inhibits Ebola membrane fusion and prevents ribonucleoprotein (RNP) release by binding to NPC1 mRNA. (**D**) PROP5 ASO binds to PDCD5 mRNA, resulting in the inhibition of programmed cell death and H1N1 influenza viral replication. (**E**) ASODN2 ASO hinders HBV replication by binding to the ASGPR1 receptor. (**F**) The binding of VIVO-PMO ASO to MRJ-L isoform results in the inhibition of HIV-1’s nuclear entry and genome integration, as well as the RSV’s viral RNA and mRNA expression. Abbreviations: ACE2, angiotensin-converting enzyme 2; miR-122, microRNA-122; NPC1, Niemann–Pick C1; PDCD5, programmed cell death protein 5; ASGPR1, asialoglycoprotein receptor 1; MRJ-L, Mammalian relative of DnaJ; HCV, Hepatitis C virus. Created in BioRender.com.

**Table 1 cells-12-02119-t001:** Antibacterial oligonucleotides.

Bacteria	ASO	Target	Ref.
*Staphylococcus aureus*	PPNA1/2	*ftsZ*	[75]
*Staphylococcus aureus*	PLNA787	*ftsZ*	[76]
*Staphylococcus aureus*	PMO	*gyrA*	[77]
*Listeria monocytogenes*	rpoA-PNA	*rpoA*	[78]
*Brucella suis*	polA-PNA	*polA*	[79]
*Acinetobacter baumannii*	AcpP-PPMO	*acpP*	[80]

**Table 2 cells-12-02119-t002:** Antiviral oligonucleotides.

Virus	ASO	Target	Ref.
Dengue virus	3′SLT PPMO	3′-SLT	[81]
Dengue virus	Vivo-MO-1	3′-SLT on 3′ UTR of genome	[82]
Respiratory Syncytial virus	AUG-2 PPMO	RSV-L mRNA	[83]
Influenza	P7-PMO	NP-v3′	[84]
Ebola	Arginine rich PPMO	VP24 mRNA	[85]
Hepatitis B virus	Ga1Nac-LNA(SSO)	HBV transcript	[86]
Human Immunodeficiency virus	FANA-ASO	HIV-1 genome	[87]

**Table 3 cells-12-02119-t003:** Antiparasitic oligonucleotides.

Parasite	ASO	Target	Ref.
*Toxoplasma gondii*	GRA10-PPMO	*GRA10*	[91]
*Trypanosoma cruzi*	Antisense 5995	*TcIP3R*	[92]
*Plasmodium falciparum*	PfCRT-VMO PfDXR-PPMO	*PfCRT* *PfDXR*	[90]
*Plasmodium falciparum*	PNA	*PfSec13*	[93]

## Data Availability

Not applicable.

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
