# Peer review of "Antisense Therapy for Infectious Diseases"

_cells, 2023, doi:10.3390/cells12162119_

Round 1
Reviewer 1 Report
IN their review, Buthelezi et al provide a nice overview over existing nucleic acid based drugs that can be used for treatment or prevention of infectious diseases. They cover viral, bacterial and parasitic diseases, which are important aspects given the recent focus on antiviral vaccines.
While the manuscript is sound on the technical side, including well-designed figures and a structure that is easy to comprehend, there are a few aspects that require significant changes to the manuscript before it can be suitable for publication.
The most important issue is the misuse of the term ASO. It is historically limited to the use where the applied oligonucleotide forms basepairs with a target nucleic acid, be it DNA or RNA. This is however not the case in two main topics covered by the authors, namely the interaction of ONs with host factors that appear to be more “aptamer”-like, i.e. in the case of the ACE2 receptor. The other nucleic-based drugs are vaccines, which are per definition not ASOs, but in most cases rather translatable sense oligos.
My suggestion would be to clarify in the necessary instances where the oligos under discussion are in fact ASOs, and where other nucleic acid-based drugs are discussed.
A second relevant issue is the discussion of delivery options. Pathogen cells have significantly different compositions and properties of their enclosing membranes, which can drastically affect delivery of the drug. This is even briefly touched when biofilms are being discussed. I believe it would be beneficial for the overall impact of this review if the current state of the art in delivery options and probably limitations of existing techniques are at least briefly discussed.
English language quality is suitable for publication.
Reviewer 2 Report
This review describes current antisense-based therapy against infectious diseases. The topic is of general interest and appealing to a wide audience because of the modern and transversal therapeutic approach described.
Overall, the manuscript is easy to read and follow, but it adds little to what is known and is somewhat confusing and misleading. The reader who is an expert in the field will not find anything new to what has already been published, nor a section on future prospects describing current and future RNA modifications that might help overcome current limitations. Conversely, the novice will be confused by the description of mRNA vaccines in an article on antisense therapy, particularly in the paragraph describing the FDA-approved antisense oligonucleotide. The mRNA vaccines are not intended for therapy, nor are they antisense. These nucleic acids do not inhibit protein expression but rather encode the protein(s) to stimulate the immune response. The abstract is also misleading in that it emphasises the threat of Mycobacterium tuberculosis and the fatal consequences that this disease claims each year. However, this bacteria is not mentioned anywhere in the article and it is not described an antisense oligonucleotide designed to counteract the replication of mycobacteria and published in 2019 (https://doi.org/10.3389/fphar.2019.01049). As it stands, the paper is unsuitable for publication in Cells.
Key Points:
1. Either delete the paragraphs describing the mRNA vaccine or create a separate section and change the title of the article accordingly. Once again, mRNA vaccines are not intended for therapy, nor are they antisense oligonucleotides;
2. Add a paragraph looking to the future describing structural modifications of antisense oligonucleotides or innovative delivery systems that could help the approach reach its full potential;
3. Describe antisense approaches against Mycobacterium spp.
Round 2
Reviewer 2 Report
The authors have duly considered the comments raised by the reviewers and the paper is suitable for publication.
For formal acceptance, however, the authors should:
1. Exchange paragraphs 6 and 7 and the accompanying figures to be consistent with the above paragraphs where bacteria were discussed before viruses;
2. Check the alignment of Table 2;
3. Use the same font for all tables. The first column of Table 3 is in a different font from Tables 1 and 2
Author Response
Response 1: We thank the reviewer for this constructive comment and have now exchanged paragraphs 6 and 7 and the accompanying figures as suggested, and it is now consistent. The sections are now as follows.
“6. Anti-mycobacterial and host factor targeted ASOs for Mycobacterium species.”
“7. ASOs targeting host factors for viral infections.”
Response 2: We have adjusted Table 2 and the table is now accurately aligned.
Response 3: We have corrected the font in Table 3 and now all the tables have the same font.